# Dissolved Oxygen and Water Temperature Drive Vertical Spatiotemporal Variation of Phytoplankton Community: Evidence from the Largest Diversion Water Source Area

**DOI:** 10.3390/ijerph20054307

**Published:** 2023-02-28

**Authors:** Zhenzhen Cui, Wanli Gao, Yuying Li, Wanping Wang, Hongtian Wang, Han Liu, Panpan Fan, Nicola Fohrer, Naicheng Wu

**Affiliations:** 1International Joint Laboratory of Watershed Ecological Security and Collaborative Innovation Center of Water Security for Water Source Region of Middle Route Project of South-North Water Diversion in Henan Province, College of Water Resource and Environment Engineering, Nanyang Normal University, Nanyang 473061, China; 2College of Water Resources, North China University of Water Resources and Electric Power, Zhengzhou 450046, China; 3Department of Hydrology and Water Resources Management, Kiel University, 24098 Kiel, Germany

**Keywords:** Danjiangkou Reservoir, phytoplankton community structure, vertical distribution, environmental factors, Surfer model

## Abstract

In order to study the vertical distribution characteristics of phytoplankton in the Danjiangkou Reservoir, the water source of the Middle Route Project of the South-North Water Diversion, seven sampling sites were set up in the Reservoir for quarterly sampling from 2017 to 2019, and water environment surveys were conducted simultaneously. The results showed that 157 species (including varieties) were identified, belonging to 9 phyla and 88 genera. In terms of species richness, Chlorophyta had the largest number of species, accounting for 39.49% of the total species. The Bacillariophyta and Cyanobacteria accounted for 28.03% and 13.38% of the total species, respectively. From the whole Danjiangkou Reservoir, the total phytoplankton abundance varied from 0.09 × 10^2^ to 20.01 × 10^6^ cells/L. In the vertical distribution, phytoplankton were mainly observed in the surface–thermospheric layer (I–II layer) and the bottom layer, while the Shannon–Wiener index showed a trend of gradually decreasing from the I–V layer. The Surfer model analysis showed that there was no significant stratification in the Q site’s water temperature (WT) and dissolved oxygen (DO) levels in the water diversion area during the dynamic water diversion process. A canonical correspondence analysis (CCA) showed that DO, WT, pH, electrical conductivity (Cond), chemical oxygen demand (COD_Mn_), total phosphorus (TP), ammonia nitrogen (NH_4_^+^-N), and total nitrogen (TN) had significant effects on the vertical distribution of phytoplankton (*p* < 0.05). A partial Mantel analysis showed that the vertical distribution of the phytoplankton community was related to WT, and the phytoplankton community structure at the other sites, except for Heijizui (H) and Langhekou (L), was affected by DO. This study has positive significance for exploring the vertical distribution characteristics of a phytoplankton community in a deep-water dynamic water diversion reservoir.

## 1. Introduction

Phytoplankton are important primary producers in water ecosystems, and their community structure and function are closely related to the water quality [1,2]. Their abundance and composition change with changes in nutrients, temperature, and light [3], and they are often used as indicator organisms for safeguarding the water quality [4]. Different environmental conditions will cause changes in the vertical distribution pattern of the phytoplankton community structure in water bodies [5,6], thus changing the water environment, which has important impacts on the structure and function of water ecosystems [4]. Therefore, a study of the vertical distribution characteristics of a phytoplankton community will be helpful in understanding a water environment comprehensively and ensure the quality of the water [7].

As a work of hydraulic engineering, a reservoir refers to an artificial lake or natural lake formed by the construction of a barrage at the mouth of the mountain or river. The complex and varied physical and chemical conditions (environment) in the reservoir constantly affect the vertical distribution of the phytoplankton community structure. For example, in the Three Gorges Reservoir, the wind and waves, turbulence intensity, and thermal stratification were the main factors affecting the vertical distribution pattern of the phytoplankton community in the Xiangxi Bay of the Three Gorges Reservoir [8,9], and an increase in the water temperature promotes the mixing of water layers and leads to a significant decrease in the phytoplankton biomass in tributaries [10]. Similarly, water temperature is an important factor affecting the vertical distribution pattern of the phytoplankton in Fuxian Lake [11]. Moreover, the reduced inflow and increased temperatures may have contributed to the abnormal proliferation of harmful algae in Lake Mead [5]. The depth of the euphotic layer is an important factor affecting the vertical distribution characteristics of the phytoplankton biomass in Qiandao Lake [12]. In Basong Cuo Lake, the vertical distribution characteristics of the phytoplankton are significantly different due to the influence of the thermocline in the dry and wet seasons [13]. In addition, the proportion of N, P, and Si nutrients is the main factor affecting the vertical distribution characteristics of the phytoplankton community structure in front of the Alcava dam in Spain [14]. In the Pacific [15] and Balliari Sea [16], summer is characterized by strong stratification and nutrient depletion, and the water body is stratified to form two layers in the euphotic zone, so that the limited nutrients in the upper layer and the limited light in the lower layer affect the phytoplankton biomass. Therefore, studying the spatial and temporal changes of the vertical distribution of the phytoplankton community structure in reservoirs will be conducive to the stability and safety of the water quality.

The Danjiangkou Reservoir is located in the city of Danjiangkou in Hubei Province and in Xichuan County in Henan Province, which borders Henan, Hubei, and Shanxi [17]. It is the water source of the Middle Route of the South–North water diversion project, which is an artificial channel with a total length of 1432 km providing water to northern China [17]. Therefore, its water quality plays an important role in the safe operation of the water diversion [17]. With the expansion of the Danjiangkou Reservoir, the rise in the water level, and the increase in storage, the quantity of exogenous nutrients brought into the basin by geological and surface runoff has increased [18]. In addition, changes in hydrological conditions and physical and chemical factors such as water level fluctuations, the hydraulic retention time, and nutrients caused by reservoir operation and management (water diversion and water supply) have significantly increased the changes in the phytoplankton community structure in the reservoir [19]. The cumulative water diversion from the Danjiangkou Reservoir over time is currently 535.41 m^3^. In previous investigations, researchers only focused on the dynamic changes in the phytoplankton communities and their driving factors, mainly concentrating on the spatial scale of the water surface [20,21,22,23,24]. The vertical spatiotemporal variation of the phytoplankton community was not considered widely. In particular, studies on the vertical variation of the phytoplankton community in the case of dynamic water diversion are very limited. In addition, because the groundwater of the central channel is transferred from the bottom of the dam, it is particularly important to study the key factors affecting the water quality on the vertical scale of different sites in the reservoir area based on the characteristics of the transferred water.

This study explores two scientific questions: (1) How does the phytoplankton community structure respond to reservoir stratification and spatiotemporal dynamics? (2) What are the key drivers among the environmental factors affecting the structure of stratified phytoplankton communities? In this study, we studied phytoplankton samples and corresponding physicochemical indices at seven different water depths during four seasons from 2017 to 2019 in the Danjiangkou Reservoir. The spatiotemporal changes in the water’s physiochemical indexes and in the phytoplankton were analyzed to reveal the main factors driving the vertical distribution of the phytoplankton. This study provides a scientific basis and data support for policy-making regarding water quality safety and the establishment of an ecological database for the Middle Route of the South-North Water Diversion project.

## 2. Materials and Methods

### 2.1. Study Area

Located in Danjiangkou City in Hubei Province and in the Xichuan County of Nanyang City in Henan Province, the Danjiangkou Reservoir has the functions of water supply, flood control, and power generation. The reservoir area is located in a subtropical monsoon climate with four distinct seasons and abundant rainfall. The Danjiangkou Reservoir, which consists of two parts, Hanku and Danku, is the largest artificial freshwater lake in Asia. The control basin covers an area of 9.5 × 10^4^ km^2^, the normal storage level is 170 m, the total storage capacity is 174.5 × 10^8^ m^3^, and the reservoir area is 1050 km^2^.

According to the geographical location characteristics of the Danjiangkou Reservoir and the degree of the impact of human activities, a total of 7 typical sampling sites were set up in the Danjiangkou Reservoir (Appendix A, Figure 1). Four sampling sites were set up in Danku: Heijizui (H), Songgang (S), Kuxin (K), and Qushou (Q). Two sampling sites were set up in Hanku: Bashang (B) and Langhekou (L). One sampling site, Taizishan (T), was set up at the junction of the Hubei and Henan provinces, and the substratum in the study area was silt. According to the actual situation (transparency = 4.1 m), seven layers (0.5 m, 5 m, 10 m, 20 m, 30 m, 50 m, and 60 m) were arranged along the vertical direction from the surface layer for sampling.

### 2.2. Sample Collection and Determination

Water and phytoplankton samples were collected in May, July, and October 2017 and in January, May, July, and October 2018–2019, respectively. The column water collector collected 2 L water samples at 0.5 m (I layer), 5 m (II layer), 10 m (III layer), 20 m (VI layer), 30 m (V layer), 50 m (VI layer), and 60 m (VII layer), respectively. After being loaded into polyethylene sampling bottles, they were stored at low temperature and protected from light for the determination of the water quality physicochemical indexes. Among them, only the water depths of B on the Han Reservoir Dam and L in the Han River exceed 50 m, and only at those sites was the VI layer of the water samples collected. The VII layer of the water at all sites was sampled according to the actual water depth. In addition, 2 L of water from the same water layer was collected by column water collector, then fixed with 30 mL of Lugo’s reagent and settled in a separator for 48 h. The supernatant was absorbed, and 30–50 mL of concentrated samples were retained for the quantitative analysis of the phytoplankton community composition.

A portable multi-parameter water quality analyzer (YSI, Professional Plus, Yellow Springs, OH, USA) was used to determine the water temperature (WT), pH, electrical conductivity (Cond), and dissolved oxygen (DO). Water transparency was measured with a 30 cm diameter Secchi disk. The depth of the euphotic zone was estimated as 2.7 times the Secchi depth [25].

The total phosphorus (TP) concentration in water was determined using ammonium molybdate spectrophotometry. The concentration of total nitrogen (TN) was determined using potassium persulfate oxidation spectrophotometry. Ammonia nitrogen (NH_4_^+^-N) was determined using Nessler’s reagent spectrophotometry, and nitrate nitrogen (NO_3_^−^-N) was determined by using ultraviolet spectrophotometry. The permanganate index (COD_Mn_) was measured using potassium permanganate digestion titration. The determination methods were chosen according to “Water and Wastewater Monitoring and Analytical Methods (Fourth edition)” [26]. An aliquot of 0.1mL of cell suspension from the cultures was taken, and 1.5 μL of Lugol’s reagent was added to fix the cells. The cell density of phytoplankton was determined using a hemocytometer (Thorma, Hirschmann, Germany) [27]. Microscope images were photographed using a camera attached to an inverted microscope (CKX41, Olympus, Japan). The phytoplankton species have been identified based on morphology [28,29,30].

### 2.3. Data Processing

The species richness index (*S*), Shannon–Wiener diversity index (*H*’), and Mcnaughton dominance index (*y*) were used to characterize the algal diversity. *S* is the number of species, and *H*’ is calculated by the formula: $H^{\prime }=-\sum P_{i}\ln P_{i}$ [31]. If *y* > 0.02, it is the dominant species, and the formula is: $y=\left(n_{i}/N\right)\;f_{i}$ [32], where $P_{i}=n_{i}/N$, $P_{i\;}$ is the ratio of the number of individuals in the ith species to the total number of all individuals, $n_{i}$ is the total number of individuals in the ith species, *N* is the number of all species, and $f_{i}$ is the frequency of occurrence of the ith species at each point.

The relevant data were collated using Excel 2016 (Microsoft, Redmond, WA, USA). The spatial distribution of the sample sites was plotted using ArcGIS 10.7 (ESRI, Redlands, CA, USA). Univariate or multivariate ANOVA was used to explore differences in the stratification or the seasonal and environmental factors within the phytoplankton communities. Pearson correlation was used to analyze the correlation between phytoplankton abundance and environmental factors. The temporal and spatial distributions of physical and chemical factors were plotted using Surfer 12 (Golden Software, Golden, CO, USA) software. Relevant statistical analyses were performed with SPSS 22 (IBM, Chicago, IL, USA) software. The principal component analysis (PCA) was used to analyze the stratified water distribution characteristics based on environmental factors. The results of the detrended correspondence analysis (DCA) showed that the length of the sorting axis was more than 3, so the canonical correspondence analysis (CCA) was used for subsequent analyses. A permutation test was performed on the ranking results of the CCA. Collinear environmental factors with a variance inflation factor (VIF) > 10 were excluded, and other environmental factors were ranked to obtain the contribution rates of the environmental factors to the vertical structure of the phytoplankton community. The bar charts of the phytoplankton species and their abundance changes were completed based on the ggplot2 package, and the variation decomposition, PCA, and CCA analyses were mainly completed based on the vegan package. A partial Mantel correlation analysis based on R (version 3.6.3) software (The R Programming Language, The University of Auckland, Auckland, New Zealand) and the software package ggcor was also completed.

## 3. Results

### 3.1. Hydrological and Environmental Characteristics of Danjiangkou Reservoir

#### 3.1.1. Principal Component Analysis of Environmental Factors

The vertical distributions of the environmental factors at different points in different years were standardized and analyzed using PCA ranking (Figure 2). The results of the PCA analysis showed that the interpretation of the PC1 axis was 97.73%, while that of the PC2 axis was 2.06%. The aggregation phenomenon was obvious in different months, and the difference was also obvious. There were also some differences in the aggregations among the vertical distributions of the sampling sites in different locations. Environmental factors differed more in time than in space, indicating that the seasonal variation of the environmental factors in the sample was more obvious, while the environmental factors in May and July were significantly different.

#### 3.1.2. Physical and Chemical Factor Analysis

From May 2017 to October 2019, the monitoring results for vertical physical and chemical factors of the Danjiangkou Reservoir showed (Table 1) that the concentrations of physical and chemical indexes showed certain characteristics of seasonal stratification changes. In the surface layer (I layer), the WT, pH, DO, and COD_Mn_ in May and July were significantly higher than those in other water layers (*p* < 0.05). The concentrations of TN and TP in the surface layer were higher than those in the bottom layer (VII layer) in May and July, and were significantly different from those in the III layer (*p* < 0.05). There was no significant difference in the vertical distribution of other parameters. There were differences in physical and chemical factors at the temporal and spatial scales during the study period.

The water temperature of the Danjiangkou Reservoir ranged from 9.6 °C to 29.5 °C, with an average of 20.7 °C. There was no significant difference among different sites during the same period. The water temperature was the highest in July, and the surface water temperature was greater than 26 °C. The maximum temperature difference was 7.6 °C in July 2018, and the water temperature was the lowest in January. The thermocline was different at different points, and the thermocline of the deep-water sample point is in the deep-water layer. The thermocline began to appear in May, and the thermocline was about 10 m. In July, the thermocline was about 15 m, and the thermal stratification disappeared after October.

In July 2018, the vertical variation of the physical and chemical factors in the Danjiangkou Reservoir was plotted using a Surfer contour model (Figure 3). The stratification of the WT and DO at the Q site in the water diversion area is not obvious compared with that of the H site on the two sides of the catchment area and the L site in the reservoir entry area, which may be related to the fact that the water diversion position of the Q site is generally located in the deep-water body, leading to water layer mixing. The changes in the concentrations of TP and COD_Mn_ in the drinking water area at the Q site were consistent with those of in the WT and DO, and the H and L sites on both sides were significantly stratified. The concentrations of NH_4_^+^-N and TN in the Danjiangkou Reservoir were low, and the stratification change was not obvious.

### 3.2. Community Characteristics of Phytoplankton Species in Danjiangkou Reservoir

#### 3.2.1. Phytoplankton Species Composition

From May 2017 to October 2019, 157 species (varieties) of 9 phyla and 88 genera of phytoplankton were identified in the Danjiangkou Reservoir (Appendix A). Chlorophyta were the most abundant phylum in terms of cell density, with 62 species (varieties) accounting for 39.49% of the total number of species. They were followed by Bacillariophyta, with 44 species (varieties) accounting for 28.03% of the total number. There were 21 species of Cyanobacteria (varieties), accounting for 13.38%. There were fewer species of Miozoa, Euglenozoa, Cryptophyta, Ochrophyta, and Charophyta, accounting for 5.10%, 4.46%, 3.82%, 3.18%, and 1.91%, respectively. Only one species of Haptophyta was found.

At the spatial scale (Figure 4a), the species richness was relatively high in the surface layer, with 119 phytoplankton species detected, followed by the II and VII layers (103 and 101 species), and the minimum number of species in the VI layer was 12. All of the sampling sites were dominated by Chlorophyta, including *Chlamydomonas* sp., *Oocystis naegelii* A.Braun and *Scenedesmus aciculatus* P.González. The Bacillariophyta were mainly *Cyclotella* sp., *Aulacoseira granulata* (Ehrenberg) Simonsen, *Fragilaria* sp., and *Synedra* sp. The numbers of species in the Danjiangkou Reservoir ranked as follows: Chlorophyta > Bacillariophyta > Cyanobacteria. At the temporal scale (Figure 4b), the Bacillariophyta–Chlorophyta algae pattern in winter and spring gradually changed to a Chlorophyta–Bacillariophyta pattern in summer and autumn. The results indicated that the phytoplankton community structure in the Danjiangkou Reservoir has spatial-temporal heterogeneity.

#### 3.2.2. Changes in Phytoplankton Abundance

The abundance of phytoplankton varied from 0.09 × 10^2^ to 20.01 × 10^6^ cells/L with a mean value of 8.14 × 10^5^ cells/L. There were obvious seasonal differences in the vertical distribution of the phytoplankton abundance (Figure 5). In I–III layers, the variation trends of dominant phyla abundance were consistent in May, and dominant phyla for the three years from 2017 to 2019 were as follows: Cyanobacteria, Chlorophyta, and Bacillariophyta. The dominant phylum of phytoplankton in VI layer changed from Cyanobacteria, to Cyanobacteria–Chlorophyta, to Bacillariophyta. The dominant phylum in VII layer changed from Bacillariophyta, to Bacillariophyta–Chlorophyta, to Bacillariophyta. In July, from 2017 to 2019, the dominant phytoplankton phyla were Cyanobacteria, Cyanobacteria–Chlorophyta, and Chlorophyta. The abundance of the dominant phytoplankton phyla in the I–III and V layers had the same changing trend. The dominant phytoplankton phyla in the VI and VII layers were Cyanobacteria and Cyanobacteria–Chlorophyta. The phytoplankton composition showed diversification in January, with no significant differences among the strata in October. Analysis of variance showed that the phytoplankton abundance varied significantly both spatially and temporally (*p* < 0.001).

#### 3.2.3. Dominant Species of Phytoplankton

A total of 18 dominant phytoplankton species (*y* > 0.02) were selected from seven sampling sites after averaging the vertical distributions of the seven layers (Table 2), and their dominance ranged from 0.020 to 0.275. *Cyclotella* sp. was the most dominant species of phytoplankton in Danjiangkou Reservoir, with a dominance of 0.275. In the I layer, the nine dominant species mainly belonged to the phyla of Chlorophyta, Bacillariophyta, and Cryptophyta, among which *Planctonema lauterbornii* Schmidle and *Cyclotella* sp. had the highest rates of dominance (0.192 and 0.186). In the VII layer, the nine dominant species mainly belonged to Chlorophyta, Bacillariophyta, and Ochrophyta, among which *Cyclotella* sp. and *Planctonema lauterbornii* had the maximum dominance (0.173 and 0.127). From the I–V layer, the dominance of *Planctonema lauterbornii* was the highest, and from the VI–VII layer, the dominance of *Cyclotella* sp. was the highest.

Two-way ANOVA of the phytoplankton diversity index showed that the total abundance, species number, and Shannon–Wiener index were significantly different in terms of the season, vertical distribution, and interactions between them (Appendix A, *p* < 0.01), except the species number. The number of species varied from 1 to 40, and the Shannon–Wiener index ranged from 0.284 to 2.771. Tukey multiple comparison showed that the total abundance was the highest in July and the lowest in January. The number of species was highest in July and October, and lowest in January. Shannon–Wiener index was the highest in July. Except for in layers I and VII, it continued to increase from January to July, reached its peak in July, and then gradually decreased, reaching its lowest point in January (Table 3). In May, compared with other vertical distributions, the Shannon–Wiener index in the surface layer decreased. The Shannon–Wiener index of the surface layer was the highest in July, compared with other months. In October, the difference was not significant. In January, the Shannon–Wiener indices of the surface and bottom layers were similar, and the other vertical distributions were not significantly different.

### 3.3. Relationship between Phytoplankton Community Structure and Environmental Factors

#### 3.3.1. Relationship between Community Structure and Environmental Factors

To clarify the relative influence of the different types of environmental factors on the community composition of the dominant phytoplankton species in the Danjiangkou Reservoir, variance decomposition was performed for the season, vertical distribution, location, and physical and chemical factors (Appendix A). These four categories of ecological factors together explained 14.51% of the variation in phytoplankton community composition, with physicochemical factors accounting for 9.51% of the variation. The second most influential factor was the seasonal factor, accounting for 2.44%. The explanation rate of the vertical distribution was 1.77%, and the explanation rate of the location was the lowest (1.51%). The combined explanation rate of the seasonal and physical and chemical factors was 1.93%. The common explanation rate of the location, vertical distribution, and physical and chemical factors was 0.10%, indicating that both seasonal variation and the spatial pattern of the sampling site led to the differences in the physical and chemical factors, which affected the distribution of the dominant phytoplankton communities.

The physicochemical factors with significant effects on the vertical distribution and composition of the phytoplankton community in the Danjiangkou Reservoir were analyzed by using partial Mantel correlation and controlling for covariates (Table 4). The results showed that the significant factors affecting the phytoplankton community were significantly different in different water layers. The results of the phytoplankton community analysis of the whole reservoir showed that the phytoplankton community formation at the H and the L sites was not affected by DO, while the phytoplankton community formation at the other sampling sites was affected by DO. The phytoplankton community composition was affected by the WT at the seven sampling sites. In addition, the formation of the phytoplankton community at the H site was significantly affected by pH, COD_Mn_, TP, NH_4_^+^-N, and NO_3_^−^-N. Similarly, the phytoplankton communities at the Q and S sites were significantly affected by COD_Mn_ (r = 0.070, *p* = 0.036 and r = 0.073, *p* = 0.034). TN was the driving factor for phytoplankton community formation at the K site (r = 0.075, *p* = 0.017). Phytoplankton community formation was significantly affected by pH and COD_Mn_ at both of the two sampling sites in the Han Reservoir. In addition, TN and NO_3_^−^-N were also significantly affected at the L site (r = 0.124, *p* = 0.001 and r = 0.124, *p* = 0.001). At the B site, the phytoplankton formation was mainly driven by TP (r = 0.068, *p* = 0.022). In the I–II layers, the formation of the phytoplankton communities at the H site and the T site was significantly affected by COD_Mn_. At the Q site, WT and NH_4_^+^-N were found to be the main drivers of phytoplankton community changes. No significant influence factors were found for the remaining sampling sites. At the III–V layers, the phytoplankton community formation was affected by DO at most sampling sites, but not at the H site and the Q site. The phytoplankton community formation was affected by the WT at all seven sampling sites, and was similar to the average of each layer of the whole reservoir. The phytoplankton communities at the L site, the B site, and the T site were significantly affected by pH. The phytoplankton community formation at the H site, the Q site, and the T site was driven by COD_Mn_. In addition, the L site was significantly affected by TN and NO_3_^−^-N, and the H site and the Q site were also significantly affected by TP. DO and WT were significant drivers of phytoplankton community changes at most sites in the water at the VI–VII layers due to the deep-water layer and the low light intensity. Phytoplankton at the T site and the B site were both driven by COD_Mn_. In addition, phytoplankton formation at the T site was significantly affected by TP, and phytoplankton formation at the B site was significantly affected by TN and NO_3_^−^-N.

#### 3.3.2. Relationship between Community Structure and Environmental Factors of Dominant Species

To further understand the influence of the environmental factors affecting the vertical distribution of the phytoplankton community, the dominant phytoplankton species and environmental factors were analyzed using CCA (Figure 6a). The results showed that DO, WT, pH, Cond, COD_Mn_, TP, NH_4_^+^-N, and TN were significantly correlated with the vertical distribution of the community (*p* < 0.05). The CCA1 axis accounted for 8.26% of the total community distribution, and the CCA2 axis accounted for 7.49% of the total community distribution. Among them, DO, pH, COD_Mn_, and TP were most closely related to the community distribution in the III–VII water layers. The WT was the most important environmental factor affecting the vertical distribution of the phytoplankton structure. Meanwhile, TP was also the main environmental factor affecting the vertical distribution. Linear regression analysis showed that the community composition was significantly correlated with the WT (r^2^ = 0.394, *p* < 0.001) and TP (r^2^ = 0.018, *p* = 0.009) in the various water layers of each site. (Figure 6b,c).

A Mantel correlation analysis was performed on the abundance of different phytoplankton species and physicochemical factors (Figure 7). The result showed that, in the whole reservoir (Figure 7a) phytoplankton community, Cyanobacteria abundance was significantly correlated with WT, pH, Cond, and COD_Mn_. Similarly, Chlorophyta abundance was significantly correlated with pH, Cond, and COD_Mn_, Bacillariophyta abundance was only associated with pH, and other phyla were associated with WT and pH. In the I–II layer (Figure 7b), Cyanobacteria abundance was correlated with the DO concentration, and Chlorophyta abundance was significantly correlated with the DO concentration and Cond. Other phyla were correlated with the DO concentration, while Bacillariophyta had little correlation with environmental factors. In the III–V layer (Figure 7c), except for Bacillariophyta, the abundance of the phytoplankton community was significantly correlated with pH. The abundance of Cyanobacteria was also related to COD_Mn_, and the abundance of Chlorophyta was also significantly related to COD_Mn_ and Cond. In the VI–VII layer (Figure 7d), the abundance of Cyanobacteria was significantly correlated with the DO concentration, the abundance of Chlorophyta was related to the DO concentration and TP concentration, and the abundance of other phyla had little correlation with environmental factors.

Pearson correlation analysis was performed between the abundance of the dominant phytoplankton species and the physicochemical factors in the whole Danjiangkou Reservoir (Figure 8a). In addition, *Chlamydomonas* sp. (r = 0.131, *p* = 0.010), *Cymbella* sp. (r = 0.115, *p* = 0.024) and *Cryptomonas erosa* (r = 0.147, *p* = 0.004) (Chlorophyta, Bacillariophyta, and Cryptophyta, respectively) were positively correlated with the DO concentration. The abundance of *Stigeoclonium* sp. (r = −0.131, *p* = 0.010) and *Radiococcus nimbatus* (De Wildeman) Schmidle (r = −0.122, *p* = 0.016) were negatively correlated with the DO concentration. Three algal species from Cyanobacteria, eight algal species from Chlorophyta, two algal species from Bacillariophyta, and three algal species from other phyla accounted for a total of sixteen algal species, and their abundance was significantly positively correlated with WT. The abundance of four species of Chlorophyta, three species of Bacillariophyta, and one species of Cryptophyta was significantly positively correlated with pH. The abundance of three algal species in Cyanobacteria, three algal species in Chlorophyta, one algal species in Bacillariophyta, and two algal species in other phyla was significantly positively correlated with COD_Mn_. A total of eight algal species, including one species from Cyanobacteria, four species from Chlorophyta, two species from Bacillariophyta, and one species from Cryptophyta, were found to be significantly positively correlated with the NH_4_^+^-N concentrations. The abundance of *Chlamydomonas* sp. (r = −0.184, *p* < 0.001) and *Cryptomonas erosa* (r = −0.114, *p* = 0.026) were negatively correlated with Cond, while the abundance of the *Dinobryon* sp. (r = 0.107, *p* = 0.036) of Ochrophyta was positively correlated with Cond. The abundance of *Planctonema lauterbornii*, *Stigeoclonium* sp., and *Chlamydomonas* sp. were significantly positively correlated with the TN and NO_3_^−^-N concentrations. The abundance of *Scenedesmus capitato-aculeatus* C.-C.Jao & Z.-Y.Hu (r = −0.102, *p* = 0.045) was negatively correlated with the TN concentration, while that of *Cryptomonas erosa* (r = 0.151, *p* = 0.003) was positively correlated with the TN concentration. The abundance of *Scenedesmus bijugus*, *Aulacoseira granulate*, and *Cymbella* sp. was negatively correlated with the NO_3_^−^-N concentration, while that of *Cryptomonas erosa* was positively correlated with the NO_3_^−^-N concentration. The abundance of *Anabaena* sp. (r = 0.204, *p* < 0.001) of Cyanobacteria was significantly positively correlated with the TP concentration.

Pearson correlation analysis was used to analyze the correlations between the abundance of the dominant phytoplankton species and the physicochemical factors at the Q site in the water diversion area (Figure 8b). The concentrations of NH_4_^+^-N in Chlorophyta were significantly positively correlated with the abundance of *Stigeoclonium* sp. (r = 0.918, *p* < 0.001), *Chlamydomonas* sp. (r = 0.861, *p* < 0.001) and *Scenedesmus capitato-aculeatus* (r = 0.905, *p* < 0.001). There was a significant negative correlation between the DO concentration and the abundance of *Aulacoseira granulata* (r = −0.410, *p* = 0.005) and *Ceratium* sp. (r = −0.319, *p* = 0.030). The *Chlamydomonas* sp. (r = 0.294, *p* = 0.047) and *Ceratium* sp. (r = 0. 303, *p* = 0.041) were significantly positively correlated with the WT. *Anabaena* sp., *Aulacoseira granulata* and *Ceratium* sp. were significantly positively correlated with COD_Mn_, while *Ankistrodesmus* sp. was significantly negatively correlated with COD_Mn_. The *Anabaena* sp. (r = 0.398, *p* = 0.006), *Aulacoseira granulata* (r = 0.418, *p* = 0.004) and *Ceratium* sp. (r = 0.351, *p* = 0.017) were positively correlated with the concentration of TP.

## 4. Discussion

### 4.1. Vertical Distribution and Succession of Phytoplankton Communities

Studies have shown that water temperature stratification is an important factor affecting the vertical distribution of phytoplankton communities. It has been found that water temperature can affect the vertical succession of the phytoplankton communities in water bodies such as Lake Mead [5], the Xin‘anjiang Reservoir [33], and the Daxi Reservoir [34]. An appropriate temperature can improve the photosynthesis of algae, accelerate the metabolism of cells, and promote their growth and proliferation [35]. The abundance of three algal species from Cyanobacteria, eight algal species from Chlorophyta, two algal species from Bacillariophyta, and three algal species from other phyla were significantly positively correlated with the WT (Figure 8a). In this study, during the reservoir mixing period in May (beginning of thermal stratification), during the dynamic water diversion process at the Q site in the water diversion area, the strong vertical mixed flow brought Bacillariophyta from the cooler water body at the bottom into the surface water body. The study showed that, in the fast-flowing high-temperature water body, the growth rate of Bacillariophyta was faster [36], which was conducive to an increase in abundance. However, the abundance of the dominant species of Bacillariophyta in the Q site of the water diversion area, *Cyclotella* sp., *Aulacoseira granulata,* and *Gomphonema* sp., had a negative correlation with the water temperature (Figure 8b), indicating that an appropriate temperature (10–15 °C) was conducive to the growth of Bacillariophyta, but an excessive increase in temperature would inhibit the abundance of Bacillariophyta. The vertical distribution of the phytoplankton’s relative abundance from the surface to the bottom layer in May 2019 was Bacillariophyta–Chlorophyta–Bacillariophyta, while the vertical distribution of phytoplankton in May 2017 was Cyanobacteria–Chlorophyta–Bacillariophyta (Figure 5). This may be due to the fact that the water temperature was not stratified in May 2019 and was affected by the vertical hydrodynamic force, while the existence of the thermocline in May 2017 led to the existence of Bacillariophyta only at the bottom of the reservoir. This is similar to the mixing periods of Fuxianhu [11], the Subtropical Reservoir [36], and the Three Gorges Reservoir [37], where the vertical distribution of phytoplankton is driven by fluid dynamics in addition to the influence of the water temperature.

Studies have shown that, in the thermal stratification period, the penetration of dissolved oxygen and nutrients in the water will be blocked by the thermocline, resulting in the oxycline and bottom hypoxia phenomena, and, at the same time, nutrients will also form a stratification phenomenon [38,39]. In this study, the number of species showed a gradually decreasing trend from the I–VI layers (Figure 4), which was related to the existence of the thermocline. However, the TN and NH_4_^+^-N concentrations were found to be uniformly distributed in the vertical direction without obvious stratification (Figure 3). This may be related to the fact that the Danjiangkou Reservoir is the Middle Route water source area, the water quality of the drinking water source area is in the oligotrophic state, and the TN and NH_4_^+^-N concentrations originally existing in the water are low. It may also be related to the fact that the existence of the thermocline does not block all material migration up and down. This study found that, in July, the water temperature increased with the increase in light, and the reservoir was in the stratification stage. The Chlorophyta were still the dominant group, but the dominant position of the Bacillariophyta decreased, and the proportion of Cyanobacteria increased. Compared with the other sites, the water diversion zone Q did not have obvious stratification during the thermal stratification period, which was related to the mixing of water layers under the strong influence of hydrodynamic force due to the dynamic water diversion process all year round. In January, with the decline in the water level, the amount of nutrients and organic matter entering the water decreased, the proportion of Cyanobacteria and Chlorophyta decreased, and the proportion of Bacillariophyta increased. Cyanobacteria, Chlorophyta, and Bacillariophyta became the common dominant species.

The increase in the nutrient concentration in the reservoir may lead to an increase in Cyanobacteria, which poses a great threat to drinking water source areas. Some studies have indicated that the Bacillariophyta phytoplankton community was at an advantage in oligotrophic water bodies [40]. This study showed that the succession of phytoplankton abundance in the I, II, and III layers from May 2017 to 2019 was as follows: Cyanobacteria—Chlorophyta—Bacillariophyta. There were significant differences in the vertical distributions of the phytoplankton communities in the Danjiangkou Reservoir. The species richness was high in the surface and bottom layers, with a decreasing trend from the I layer to the VI layer, explained by the fact that the nutrient level in the Danjiangkou Reservoir water decreased year by year from 2017 to 2019. Wang Yinghua [41] et al. found that Bacillariophyta were the dominant group in the Danjiangkou Reservoir in spring and winter from 2014 to 2015, but that Chlorophyta was dominant in summer. In this study, we found that the succession of phytoplankton abundance in the I, II, III, and VII layers from July 2017 to 2019 was as follows: Cyanobacteria—Cyanobacteria and Chlorophyta—Chlorophyta. Our results were more similar to those of Wang Yinghua and may be related to the expansion of the Danjiangkou Reservoir dam in 2012, which affected the material and energy exchange of the Danjiangkou Reservoir, thereby changing the seasonal dynamics of the phytoplankton community. Feng Yumo [42] found that, after the expansion of Haohanpo Reservoir, the abundance and biomass of phytoplankton increased, among which Chlorophyta and Cyanobacteria increased the most obviously. Li Xiaoyan [43] found that the slow flow of water in the Manwan Reservoir after dam construction resulted in nitrogen and phosphorus accumulation, which was conducive to the growth of Chlorophyta.

The Shannon–Wiener index showed a decreasing trend from the I–V layers in January, July, and October. The decrease in species diversity implied the simplification of the ecological structure, indicating that the simplification of the ecological structure would affect the productivity and stability of the ecosystem as the water depth increased. The rise in the Shannon–Wiener index in the seventh layer may be related to the fact that the sampling reaches the sediment. The bottom layer is anoxic, but it will promote the release of soluble phosphorus [44], which will provide nutrients for the growth and reproduction of phytoplankton.

### 4.2. Effects of Environmental Factors on Phytoplankton Communities

Although previous studies have shown that there are differences in the phytoplankton communities and the dominant species abundance in stratified water bodies, the phytoplankton abundance in the surface layer and the thermocline is mainly affected by environmental variables such as water temperature and light [45,46]. However, the understanding of the spatial heterogeneity of the influencing factors of the phytoplankton community is still insufficient, which limits the understanding and effective management of the ecosystem function of the Danjiangkou Reservoir. This result showed that, in addition to physical factors such as pH and WT, the DO, COD_Mn_, TN, TP, NH_4_^+^-N, and TN concentrations had significant effects on phytoplankton community dynamics, and the important drivers of the phytoplankton community structure varied greatly among the sites. For example, the water diversion area Q of the Middle Route project showed a large difference from the other ecological zones, and the phytoplankton community structure of the I–II layers were mainly strongly correlated with the concentration of NH_4_^+^-N, while III–V layers were related to the TP concentration (Table 4). NH_4_^+^-N is the main nitrogen source for phytoplankton, and TP is the available nutrient for phytoplankton, which provides opportunities for phytoplankton to grow. In addition, NH_4_^+^-N is mostly emitted from human activities [47], while TP may come from the accumulation of nutrients in the thermocline and the release of soluble phosphorus from the sediment. Moreover, the higher concentration of NH_4_^+^-N and TP brought by the Han River provides more bioavailable nitrogen and phosphorus sources for the drinking water areas [48], which will aggravate the risk of eutrophication and algal bloom. The Pearson correlation analysis (Figure 8b) showed that an increasing NH_4_^+^-N concentration led to an increase in the abundance of the phytoplankton Chlorophyta. There was a significant positive correlation between the NH_4_^+^-N concentration and the concentrations of *Stigeoclonium* sp., *Chlamydomonas* sp., and *Scenedesmus capitato-aculeatus*. Increasing the TP concentration led to an increase in the phytoplankton biomass, with the abundance of *Anabaena* sp., *Aulacoseira granulata* and *Ceratium* sp. all being positively correlated with the TP concentration. The difference between NH_4_^+^-N and other ecological zones in the water diversion zone may be related to the mixing of water layers due to the fact that the diversion position is generally located a deep water body during the dynamic water diversion. The Danjiangkou Reservoir is located in the north subtropical zone and warm temperate zone. It is affected by the subtropical monsoon climate and has four distinct seasons, with obvious temperature differences in different seasons. Heavy rainfall in summer and autumn led to a significant seasonal variation in the inflow and outflow and an increased disturbance in the upstream inflow to the whole Danjiangkou Reservoir. Therefore, the abundance of the phytoplankton in the Danjiangkou Reservoir is higher in summer, when the temperature is relatively high [8]. The phytoplankton community at the Dan reservoir entry site H and the Han reservoir entry site L were significantly driven by nutrients. Therefore, summer is a critical season for phytoplankton control, and it is necessary to monitor incoming runoff organic matter and possible pollution sources. A partial Mantel analysis (Table 4) showed that the phytoplankton community in the I–II layers of the water was not affected by nutrients except for Q in the water diversion area. In addition to nutrients, water level fluctuations and meteorological conditions are important driving factors for phytoplankton community changes in reservoirs [49]. Therefore, studies of the driving factors of the phytoplankton community in the Danjiangkou Reservoir may require more consideration of hydrological and meteorological factors, as well as of water level fluctuations caused by water diversions.

Studies have shown that the phytoplankton community structure changes with the availability of nutrients and light [3]. Nitrogen and phosphorus are the nutrients that drive the vertical distribution of the phytoplankton community structure in front of the Alcava dam in Spain [14]. The radiolucent zone formed by water stratification has limited nutrients in the upper layer and limited light in the lower layer, which then affects the phytoplankton biomass in the Pacific [15] and in the Baliari Sea [16]. Wind and waves, turbulence intensity, and thermal stratification affect the vertical distribution pattern of the phytoplankton community in the Xiangxi Reservoir Bay of the Three Gorges Reservoir [8,9].

In the I–II water layers, the abundance of *Chlamydomonas*, the dominant species of Chlorophyta, was significantly and positively correlated with the DO concentration (Figure 7), because there was sufficient dissolved oxygen, suitable light, and a suitable temperature for phytoplankton growth. The abundance of *Chlamydomonas* sp., *Cymbella* sp. and *Cryptomonas erosa* was also significantly positively correlated with the DO concentration. These results indicate that phytoplankton require sufficient oxygen for respiration and can also release oxygen through photosynthesis, acting as a regulator of the dissolved oxygen content in the water. Thermocline formation separates two distinct environments with different resource constraints (dissolved oxygen, nutrients, and light), thereby regulating the phytoplankton community structure [50]. In the VI–VII water layers, the abundance of *Stigeoclonium* sp. and *Radiococcus nimbatus* was negatively correlated with the DO concentration. Under hypoxic conditions, iron hydroxide in the overlying water can combine with phosphorus and displace it into ferrous ions, while releasing phosphorus from the sediment [51]. Bottom hypoxia can promote the endogenous release of soluble reactive phosphorus and provide the nutrients required for phytoplankton growth and reproduction. In this study, the phytoplankton community was strongly affected by water stratification [52], which is similar to the findings of previous studies.

Below the thermocline, nutrients and light availability strongly modulate the composition of the community [53]. The components of the phytoplankton community have shown differences in their nutrient uptake efficiency and requirements [54] and in their light quantity and quality adaptation [55]. Studies on vertical zooplankton distribution showed that vertical stratification can hinder the migration of some small zooplankton groups and suggests different grazing pressures above and below the thermocline [56], considering that zooplankton filter feeders and heterotrophic flagellates are the main grazers of the predominant picophytoplankton during the summer stratification [57]. Studies of the phytoplankton community composition between the upper and lower water layers of the thermocline should focus on whether the differences in phytoplankton predation pressure are related to different predators, which was not considered in this study.

## 5. Conclusions

The overall results indicated that the phytoplankton community structure in the Danjiangkou Reservoir had obvious spatial and temporal heterogeneity, with significant differences in the vertical and seasonal phytoplankton community structure. Water temperature, DO, and nutrients were the main drivers of the vertical phytoplankton distribution in the reservoir. There were significant differences in the environmental factors of the Danjiangkou Reservoir in different seasons, which affected the phytoplankton community structure. During the stratification period (July), the thermocline blocked material migration, but the TN and NH_4_^+^-N concentration in the oligotrophic state of the drinking water source area were not affected, and the phytoplankton species richness was higher than that in other seasons. Water layers mixed during the dynamic water diversion at the Q site in the diversion area, and the phytoplankton community in the I–II layers was significantly driven by the NH_4_^+^-N concentration, while that in the III–V layers was significantly driven by the TP concentration. There were great differences in the important driving factors of the vertical distribution of the phytoplankton community among different sites, suggesting that ecological zoning management should be carried out for the Danjiangkou Reservoir.

## Figures and Tables

**Figure 1 ijerph-20-04307-f001:**
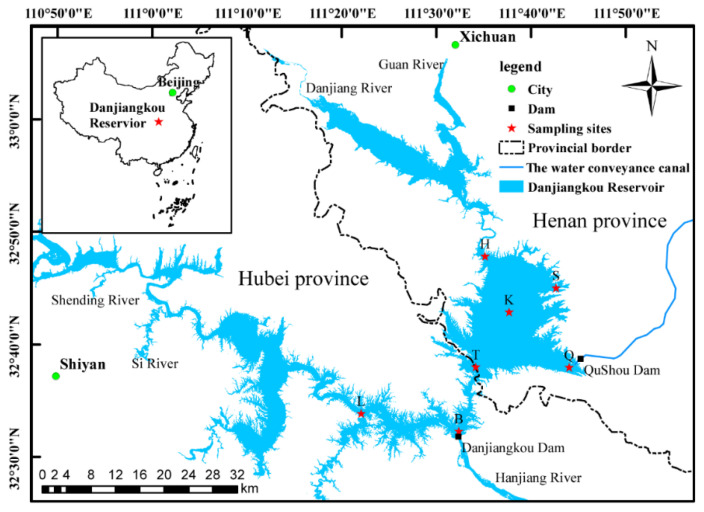
Sketch map of the 7 sampling sites in the Danjiangkou Reservoir.

**Figure 2 ijerph-20-04307-f002:**
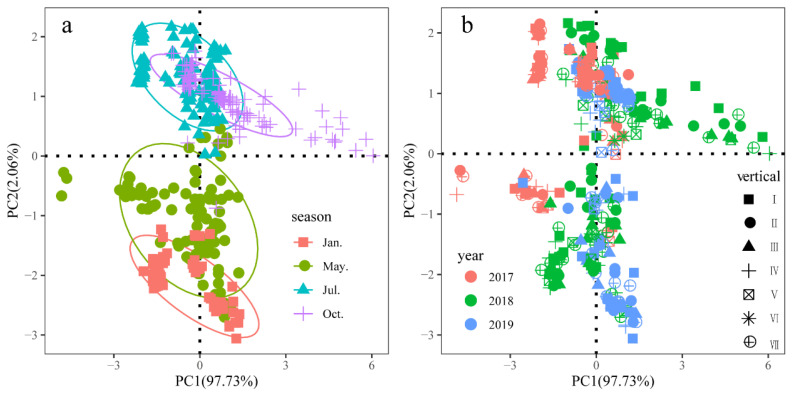
PCA of vertical water environmental factors in the Danjiangkou Reservoir ((**a**): Season; (**b**) Vertical).

**Figure 3 ijerph-20-04307-f003:**
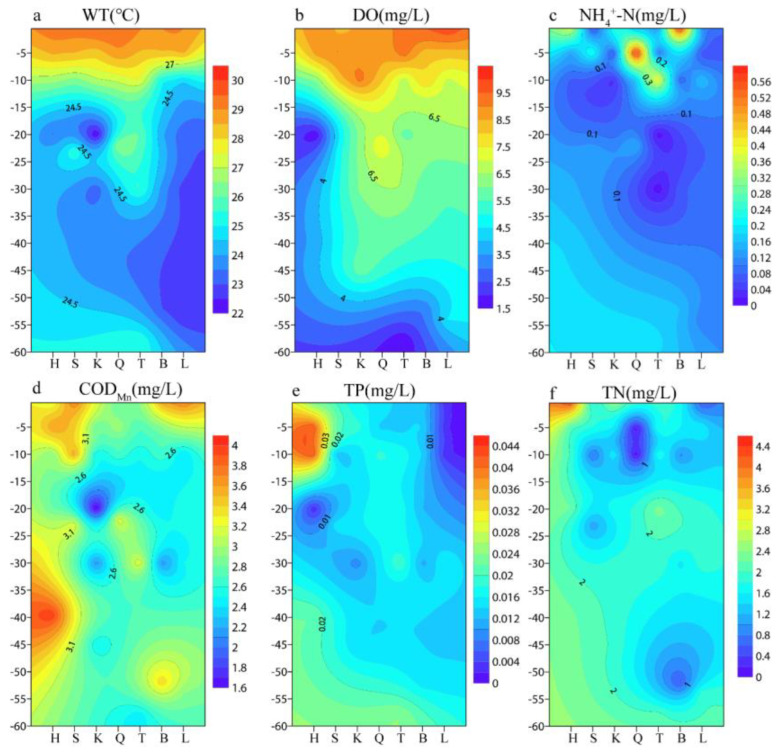
Variation of vertical physicochemical parameters in the Danjiangkou Reservoir ((**a**) WT; (**b**) DO; (**c**) NH_4_^+^-N; (**d**) COD_Mn_; (**e**) TP; (**f**) TN).

**Figure 4 ijerph-20-04307-f004:**
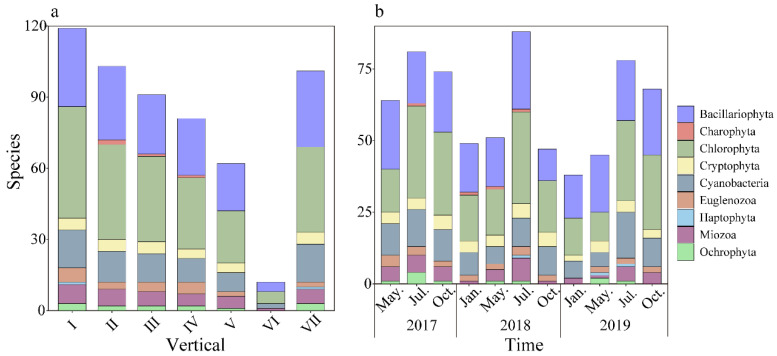
Histogram of phytoplankton species composition in the Danjiangkou Reservoir ((**a**) vertical; (**b**) time; the Roman numerals were the layers in the water column).

**Figure 5 ijerph-20-04307-f005:**
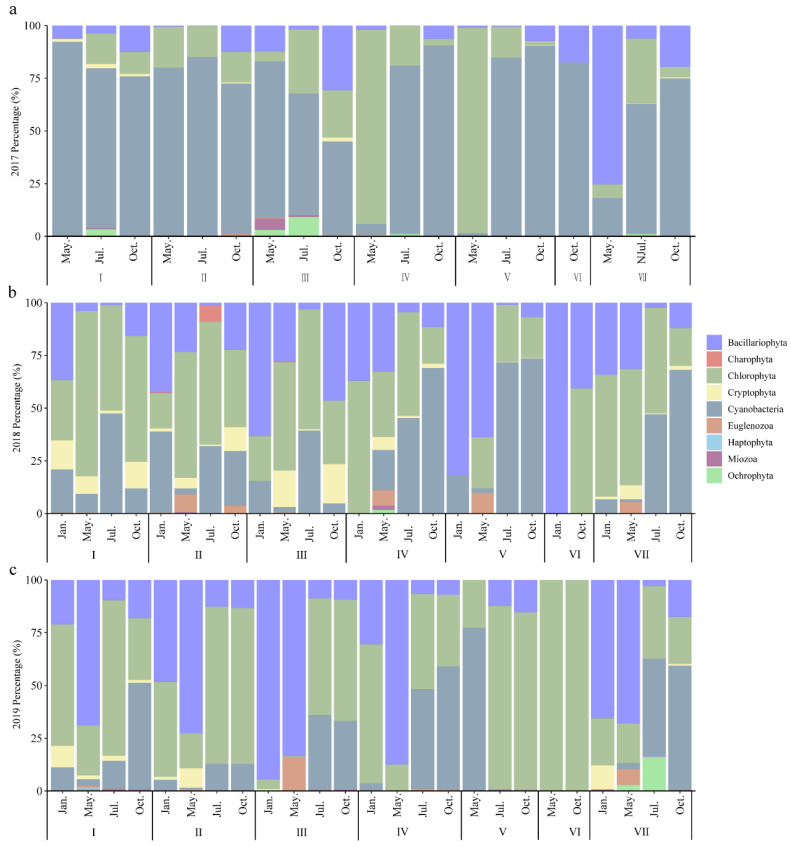
Relative abundance of vertical phytoplankton in the Danjiangkou Reservoir ((**a**) 2017 percentage; (**b**) 2018 percentage; (**c**) 2019 percentage; the Roman numerals were the layers in the water column).

**Figure 6 ijerph-20-04307-f006:**
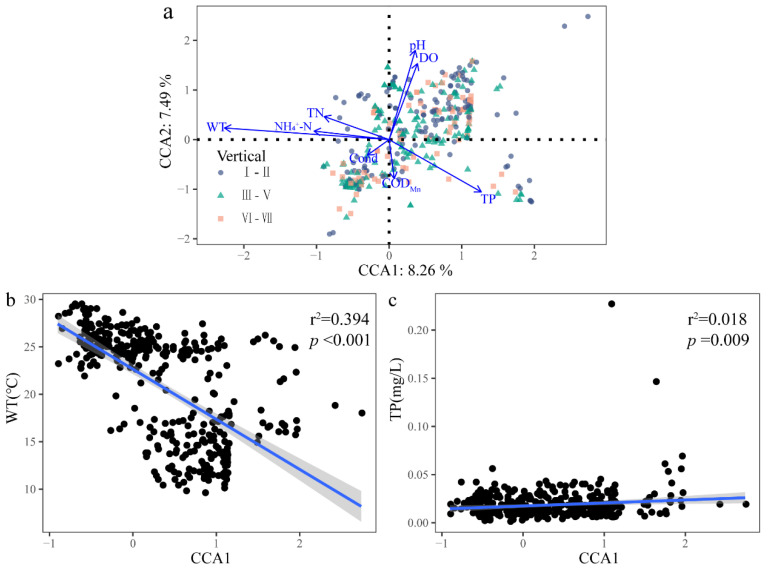
CCA of phytoplankton communities and physicochemical factors in the Danjiangkou Reservoir. ((**a**) CCA ordination; (**b**) Linear regression of DO with CCA1; (**c**) Linear regression of TP versus CCA1).

**Figure 7 ijerph-20-04307-f007:**
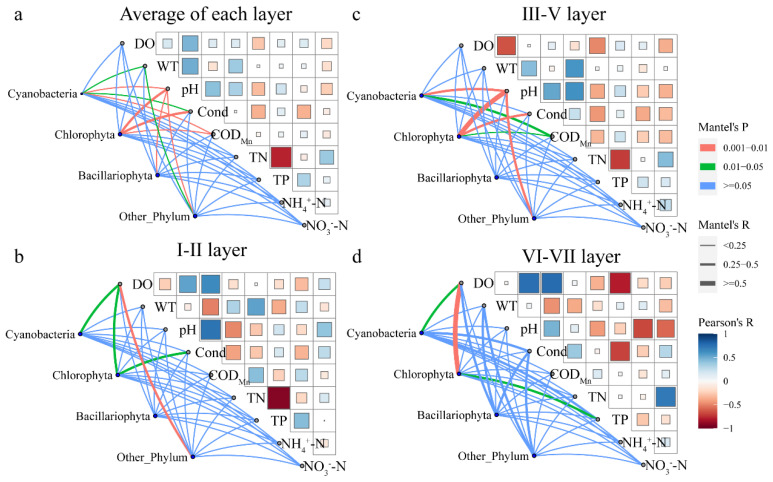
Physicochemical variables and abundance of phytoplankton phyla composition in the Danjiangkou Reservoir. ((**a**) Average of each layer; (**b**) I–II layer; (**c**) III–V layer; (**d**) VI–VII layer).

**Figure 8 ijerph-20-04307-f008:**
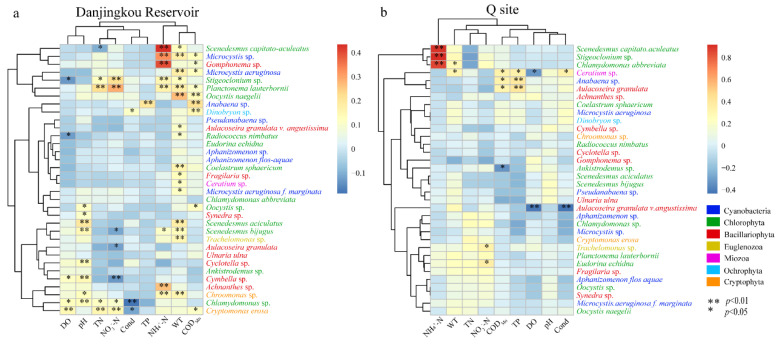
Heatmap of correlation between phytoplankton dominant species abundance and environmental parameters. ((**a**) Danjiangkou Reservoir; (**b**) Q site).

**Table 1 ijerph-20-04307-t001:** Variation of vertical water quality factor parameters in the Danjiangkou Reservoir.

Parameters	I Layer	II Layer	III Layer	IV Layer	V Layer	VI Layer	VII Layer
WT (°C)	21.44 ± 6.08 a	21.09 ± 6.06 ab	20.71 ± 5.70 ab	20.32 ± 5.54 ab	21.10 ± 5.11 ab	20.98 ± 5.22 ab	19.28 ± 5.81 b
pH	8.25 ± 0.44 a	8.21 ± 0.44 ac	8.14 ± 0.45 ad	8.08 ± 0.43 bcd	7.98 ± 0.43 bd	7.80 ± 0.53 b	8.07 ± 0.48 bcd
DO (mg/L)	8.98 ± 1.67 a	9.00 ± 2.22 a	8.55 ± 2.11 ac	7.82 ± 2.42 bc	7.34 ± 1.86 b	6.84 ± 2.02 b	7.18 ± 2.56 b
Cond (μS/cm)	265.82 ± 41.12	264.63 ± 41.15	266.12 ± 41.34	269.91 ± 42.49	263.04 ± 35.73	259.63 ± 15.43	266.94 ± 43.52
NO_3_^−^-N (mg/L)	0.937 ± 0.378	0.916 ± 0.278	0.908 ± 0.284	0.923 ± 0.321	0.983 ± 0.295	0.978 ± 0.189	0.931 ± 0.345
NH_4_^+^-N (mg/L)	0.068 ± 0.084	0.070 ± 0.080	0.070 ± 0.071	0.052 ± 0.037	0.055 ± 0.031	0.068 ± 0.067	0.061 ± 0.043
TN (mg/L)	1.30 ± 0.55 a	1.18 ± 0.48 ab	1.15 ± 0.44 bc	1.32 ± 0.40 a	1.32 ± 0.41 ac	1.27 ± 0.35 ac	1.28 ± 0.41 ac
TP (mg/L)	0.018 ± 0.012	0.018 ± 0.010	0.018 ± 0.011	0.018 ± 0.010	0.017 ± 0.009	0.019 ± 0.010	0.018 ± 0.010
COD_Mn_ (mg/L)	2.82 ± 0.59 a	2.71 ± 0.52 ab	2.63 ± 0.41 b	2.63 ± 0.54 b	2.52 ± 0.47 b	2.44 ± 0.24 b	2.66 ± 0.53 b

Note: Values were arithmetic means ± standard deviations. Different lowercase letters represented significant differences by ANOVA at *p* < 0.05. Means with the same letter were not significantly different.

**Table 2 ijerph-20-04307-t002:** Dominant species and dominant degree of phytoplankton in the Danjiangkou Reservoir.

Phylum	Dominant Species	I	II	III	IV	V	VI	VII
Cyanobacteria	*Aphanizomenon* sp.	-	-	0.033	-	-	-	-
	*Microcystis aeruginosa* Kützing	-	-	-	-	0.029	-	-
Chlorophyta	*Chlamydomonas* sp.	0.130	0.105	0.074	0.103	0.100	0.169	0.121
	*Eudorina echidna* Svirenko	-	-	-	-	-	0.025	-
	*Oocystis naegelii*	0.040	0.031	0.039	0.045	0.050	0.089	0.046
	*Planctonema lauterbornii*	0.192	0.229	0.260	0.213	0.224	-	0.127
	*Scenedesmus* sp.	-	-	-	-	-	-	-
	*S. aciculatus*	-	0.021	0.033	-	-	0.039	-
	*S. bijugus* (Turpin) Lagerheim	-	0.023	-	0.020	0.042	-	-
	*Stigeoclonium* sp.	-	0.022	0.024	-	-	-	0.022
Bacillariophyta	*Aulacoseira granulata*	0.109	0.117	0.140	0.148	0.116	0.184	0.126
	A. var. *angustissima* (O.Müller) Simonsen	0.040	0.037	0.042	0.050	0.061	0.181	0.046
	*Cyclotella* sp.	0.186	0.177	0.170	0.201	0.118	0.275	0.173
	*Fragilaria* sp.	0.036	0.022	0.034	0.049	0.040	-	0.061
	*Ulnaria ulna* (Nitzsch) Compère	-	-	-	-	0.038	-	-
Ochrophyta	*Dinobryon* sp.	-	0.022	0.028	-	-	-	0.021
Cryptophyta	*Chroomonas* sp.	0.021	0.025	0.023	-	-	-	-
	*Cryptomonas erosa* Ehrenberg	0.021	-	-	-	-	-	-

Note: “-” indicates that this species appears during the study period but was not the dominant species.

**Table 3 ijerph-20-04307-t003:** Differences in vertical distribution of Shannon–Wiener indices of phytoplankton in the Danjiangkou Reservoir in different seasons.

Layer	Jan.	May.	Jul.	Oct.
I	1.795 ad	1.011 bd	2.117 a	1.997 af
II	0.998 be	1.346 bd	1.87 a	1.95 af
III	0.755 be	1.128 bd	1.827 a	1.786 a
IV	0.607 ce	1.249 bd	1.846 a	1.477 ab
V	0.366 be	1.288 ad	1.64 a	1.324 a
VI	0.693 de	/	/	0.901 f
VII	1.272 bd	1.223 bd	1.785 a	1.734 af

Note: Values were arithmetic averages of the samples. Different lowercase letters in the same line represented significant differences at the *p* < 0.05 level. Means with the same letter were not significantly different.

**Table 4 ijerph-20-04307-t004:** Mantel test of phytoplankton community and physical and chemical factors in different water layers in the Danjiangkou Reservoir.

Water Layer		H	S	K	Q	T	B	L
I–II layers	DO	0.031	0.081	−0.015	0.051	0.056	0.005	0.013
	WT	−0.031	0.084	−0.008	**0.123 ***	0.058	0.096	0.061
	pH	−0.010	−0.086	−0.016	−0.031	−0.038	0.058	−0.027
	Cond	−0.089	−0.092	−0.038	−0.051	−0.014	−0.117	−0.081
	COD_Mn_	**0.153 ***	0.029	−0.038	0.045	**0.158 ***	0.007	−0.019
	TN	−0.038	0.047	0.013	0.076	−0.118	−0.144	−0.008
	TP	0.078	−0.040	−0.030	−0.054	0.103	0.006	0.070
	NH_4_^+^-N	0.018	0.089	−0.019	**0.158 ****	−0.049	−0.014	0.027
	NO_3_^−^-N	−0.033	−0.018	−0.059	−0.096	−0.205	−0.038	−0.076
III–V layers	DO	0.082	**0.267 ***	**0.187 ****	0.120	**0.269 ****	**0.159 ****	**0.211 ****
	WT	**0.142 ***	**0.442 ****	**0.240 ****	**0.216 ***	**0.297 ****	**0.249 ****	**0.138 ***
	pH	0.095	−0.109	0.057	−0.082	**0.116 ***	**0.123 ***	**0.119 ***
	Cond	−0.004	−0.085	0.070	−0.077	0.053	−0.019	0.068
	COD_Mn_	**0.142 ***	0.045	−0.072	**0.214 ***	**0.122 ***	−0.005	−0.018
	TN	0.106	−0.027	0.092	0.005	−0.026	−0.020	**0.187 ****
	TP	**0.195 ****	0.116	−0.077	**0.226 ***	0.014	0.072	−0.010
	NH_4_^+^-N	−0.034	0.014	0.034	0.128	0.031	0.025	−0.035
	NO_3_^−^-N	0.097	−0.076	0.041	−0.062	−0.030	0.029	**0.129 ***
VI–VII layers	DO	0.017	**0.295 ****	0.069	**0.323 ***	0.097	**0.211 ***	0.002
	WT	0.110	**0.538 ****	**0.357 ***	**0.523 ****	**0.450 ***	**0.218 ***	**0.325 ****
	pH	0.304	−0.061	0.001	−0.146	0.181	0.041	−0.067
	Cond	0.312	0.035	−0.084	0.029	−0.122	−0.005	−0.024
	COD_Mn_	0.285	−0.073	−0.145	0.070	**0.232 ***	**0.220 ***	0.096
	TN	−0.148	0.087	−0.095	−0.091	−0.072	**0.281 ****	0.065
	TP	0.070	0.015	0.023	0.072	**0.212 ***	0.098	0.077
	NH_4_^+^-N	0.153	0.020	0.036	−0.201	−0.067	0.080	−0.176
	NO_3_^−^-N	0.055	−0.240	0.060	−0.048	−0.139	**0.378 ****	0.176
Average of each layer	DO	0.035	**0.179 ****	**0.062 ***	**0.143 ****	**0.170 ****	**0.080 ****	0.011
	WT	**0.094 ****	**0.306 ****	**0.188 ****	**0.223 ****	**0.268 ****	**0.204 ****	**0.168 ****
	pH	**0.091 ***	0.019	0.024	−0.060	**0.072 ***	**0.114 ****	**0.058 ***
	Cond	0.041	0.034	0.056	−0.019	0.003	0.010	0.033
	COD_Mn_	**0.107 ****	**0.073 ***	−0.021	**0.070 ***	**0.186 ****	**0.066 ***	**0.072 ***
	TN	0.060	0.022	**0.075 ***	−0.016	−0.003	0.032	**0.124 ****
	TP	**0.144 ****	0.061	0.013	0.042	0.039	**0.068 ***	0.046
	NH_4_^+^-N	**0.083 ***	−0.002	0.031	0.046	0.009	0.053	−0.011
	NO_3_^−^-N	**0.072 ***	0.019	0.051	0.042	−0.027	0.039	**0.124 ****

Note: Significant factors were indicated in bold font, * denotes *p* < 0.05 and ** denotes *p* < 0.01.

## Data Availability

Not applicable.

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
