# Peer review of "Dissolved Oxygen and Water Temperature Drive Vertical Spatiotemporal Variation of Phytoplankton Community: Evidence from the Largest Diversion Water Source Area"

_ijerph, 2023, doi:10.3390/ijerph20054307_

Round 1

Reviewer 1 Report

The manuscript is devoted to studying the influence of important environmental factors (mainly temperature and dissolved oxygen) on the vertical distribution of structural parameters of phytoplankton in the Danjiangkou Reservoir. Such studies are of great interest, especially in the context of understanding the water environment comprehensively and ensuring the health of water body. The manuscript was well written.

Some points require clarification.

   1.  What was the rationale for the choice of depths from which samples were taken? Why was transparency not defined? And according to the transparency - the depth of the photic zone? This would be a more reliable way to select the sampling depth.

22.  It is not clear how the number of phytoplankton cells was counted, since it is not indicated what volume of sample was added to the chamber for counting phytoplankton. What was used as a unit for measuring the abundance of colonial forms of algae - a single cell or a colony? How was the number of cells counted in large colonies, such as Microcystis?

Some minor corrections that should be made to the paper are listed here.

a.      In figures 5 and 6 it is necessary to give an explanation that the Roman numerals are the layers in the water column.

b.      The title to figure 5 indicates about Circle diagram, and the figure itself shows column diagrams.

c.      At the first mention of a species of algae, it is necessary to indicate its name with the author, for example Scenedesmus aciculatus P.González

All the best

Author Response

Thanks very much for taking your time to review this manuscript. We really appreciate all your comments and suggestions. Authors of the manuscript (Manuscript ID: ijerph-2214086) have made revisions based on your suggestions.

The manuscript is devoted to studying the influence of important environmental factors (mainly temperature and dissolved oxygen) on the vertical distribution of structural parameters of phytoplankton in the Danjiangkou Reservoir. Such studies are of great interest, especially in the context of understanding the water environment comprehensively and ensuring the health of water body. The manuscript was well written.

Some points require clarification.

  1.  What was the rationale for the choice of depths from which samples were taken? Why was transparency not defined? And according to the transparency - the depth of the photic zone? This would be a more reliable way to select the sampling depth.

Reply: Thanks for your kind comments. We have revised the sentence as follows (lines 124, 145-147): “Water transparency was measured with a 30 cm diameter Secchi disk. Depth of euphotic zone was estimated as 2.7 times the Secchi depth.”

  1. It is not clear how the number of phytoplankton cells was counted, since it is not indicated what volume of sample was added to the chamber for counting phytoplankton. What was used as a unit for measuring the abundance of colonial forms of algae - a single cell or a colony? How was the number of cells counted in large colonies, such as Microcystis?

    Reply: Thanks for your kind reminders. We have revised the sentence as follows (lines 155-160): “An aliquot of 0.1mL of cell suspension from the cultures was taken and 1.5 μL of Lugol’s reagent was added to fix the cells. The cell density of phytoplankton was determined using a hemocytometer (Thorma, Hirschmann, Germany) [27]. Microscope images were photographed using a camera attached to an inverted microscope (CKX41, Olympus, Japan). Phytoplankton species have been identified based on morphology [28-30].”

Some minor corrections that should be made to the paper are listed here.

  1. In figures 5 and 6 it is necessary to give an explanation that the Roman numerals are the layers in the water column.

Reply: Thanks for your suggestion. We have revised the explanation in figures 4 and 5 (lines 265, 284-285).

  1. The title to figure 5 indicates about Circle diagram, and the figure itself shows column diagrams.

Reply: Thanks for your kind reminders. We have revised the caption of Figure 4 (line 264).

  1. At the first mention of a species of algae, it is necessary to indicate its name with the author, for exampleScenedesmus aciculatus P.González

Reply: Thanks for your kind reminders. We have revised the Latin name of the algae.

Reviewer 2 Report

The studies described and analyzed in this manuscript are undeniably very interesting. However, the way they are described does not give a complete picture of the results achieved.

Detailed comments

Introduction

- the phrase "water health" (line 47) and "health of water body" (line 53) are used - this phrase does not seem to be the most appropriate.

Materials and Methods

- please unify the nomenclature - the text uses the phrase "sampling sites" (line 119), in the description of Figure 1 "sampling point", and in the Supplement "monitoring sites"

- I don't really understand what samples were taken at which point. From the information in the text and in the Supplement, it appears that the depth at every point was not such as to collect samples called the VI layer or VII layer. In addition, the average depth of the points is given in the Supplement. Shouldn't it be the maximum depth. Such a misunderstanding about the depth and the samples taken from each point affects the fact that I cannot properly read the results described later.

- please note the use of the software name Surfer - Sufer is in the text.

Results

This chapter needs a major rewrite.

- Figure 2b - it is hard to read

 Subchapter 3.2.1 - the text describes the results for specific months and years, and the Authors refer to Table 1, in which the averages for the entire research period are given. It may be better to provide accurate results, for example in a Supplement.

- Figure 3 - the graphs show that at each research point, samples from all 7 layers were taken. How does this relate to the depth of these points, have samples from each layer been taken at each point (see my note to the Materials and Methods chapter).

- Figure 4 - Is this graph necessary? It is very impressive but adds little to the text.

- Figure 5 - wrong caption - it's not a circle diagram

Subchapter 3.2.3 - descriptions are incomprehensible in relation to what is presented in table2. There are references to layers in the text, which unfortunately I can't see in any chart or table.

I also do not understand the calculations in the text, e.g. In May, the 12 dominant species were mainly belonging to the phyla of Bacillariophyta and Chlorophyta, among which Cyclotella sp. and Chlamydomonas sp. had higher dominance (0.244 and 287 0.238) (lines 284-287 ). The table shows that there were 8 dominant species. This is the case in all groups of algae given in the text.

Subchapter 3.3.1. – why descriptions of results in this subchapter are suddenly for combined layers and not for each separately, as was the case in previous subchapters.

Why is the Pearson corellation analysis (lines from 442) separate for point Q. On what basis the authors decided to present this analysis separately for this point, since the other analyzes for this point are not separated.

All these inaccuracies in the presentation of the results (once for layers, once for connected layers, once for a single point) make the results incomprehensible. They require redrafting, then the discussion will also have substantive value.

General comments

There are some mistakes in the Latin names - in table 2 - it should be Ankistrodesmus instead of Ankistrodemus. The text says Aulacoseira granulate instead of Aulacoseira granulata (lines 438 and 475).

There are some mistakes in using spaces.

Author Response

Thanks very much for taking your time to review this manuscript. We really appreciate all your comments and suggestions. Authors of the manuscript (Manuscript ID: ijerph-2214086) have made revisions based on the suggestions.

Introduction

- the phrase "water health" (line 47) and "health of water body" (line 53) are used - this phrase does not seem to be the most appropriate.

Reply: Thanks for your kind reminders. We have revised the phrase - ”water quality safeguard” (line 47) and “quality of water safeguard” (line 53).

Materials and Methods

- please unify the nomenclature - the text uses the phrase "sampling sites" (line 119), in the description of Figure 1 "sampling point", and in the Supplement "monitoring sites"

Reply: Very sorry for this mistake. We have named the sampling sites uniformly in this paper as sampling sites (line 128).

- I don't really understand what samples were taken at which point. From the information in the text and in the Supplement, it appears that the depth at every point was not such as to collect samples called the VI layer or VII layer. In addition, the average depth of the points is given in the Supplement. Shouldn't it be the maximum depth. Such a misunderstanding about the depth and the samples taken from each point affects the fact that I cannot properly read the results described later.

Reply: Yes, you are right. Our thoughtlessness causes dyslexia to our readers. We have changed the average water depth of each sampling site provided in the supplementary file to the maximum depth as a reference. The B and L sites collected samples from 50m called the â…¥ layer (lines 135-137). All sampling sites collected samples from the actual maximum depth called the â…¦ layer (lines 137-138).

- please note the use of the software name Surfer - Sufer is in the text.

Reply: Sorry for this mistake. We have revised the software name Surfer - Sufer in the text.

Results

This chapter needs a major rewrite.

- Figure 2b - it is hard to read

Reply: Yes, we agree with you. We also think it was difficult to understand, because there were too many indicators, the clustering was not obvious enough, and the confidence intervals overlap too much, which would be confusing if added.

 Subchapter 3.2.1 - the text describes the results for specific months and years, and the Authors refer to Table 1, in which the averages for the entire research period are given. It may be better to provide accurate results, for example in a Supplement.

Reply: Thanks for your suggestion. We have added the specific data in Supplemental table 2.

- Figure 3 - the graphs show that at each research point, samples from all 7 layers were taken. How does this relate to the depth of these points, have samples from each layer been taken at each point (see my note to the Materials and Methods chapter).

Reply: Thanks for your comments. Samples from the â… -â…¤ and â…¦ layers were collected at each sampling site, and samples from layer â…¥ were collected at the B and L sites. The rest of the data were calculated by Kriging interpolation with Surfer software. The sampling depth of a sample was defined as layers.

- Figure 4 - Is this graph necessary? It is very impressive but adds little to the text.

Reply: Thanks for your kind reminders. We have deleted Figure 4.

- Figure 5 - wrong caption - it's not a circle diagram

Reply: Very sorry for this mistake. We have revised the caption of Figure 4 (line 264).

Subchapter 3.2.3 - descriptions are incomprehensible in relation to what is presented in table2. There are references to layers in the text, which unfortunately I can't see in any chart or table.

Reply: Yes, we agree with you. We have reprocessed the data in Table 2 (line 298).

I also do not understand the calculations in the text, e.g. In May, the 12 dominant species were mainly belonging to the phyla of Bacillariophyta and Chlorophyta, among which Cyclotella sp. and Chlamydomonas sp. had higher dominance (0.244 and 287 0.238) (lines 284-287 ). The table shows that there were 8 dominant species. This is the case in all groups of algae given in the text.

Reply: Thanks for your suggestion. We have modified the description of 3.2.3 according to Table 2 (lines 287-297).

Subchapter 3.3.1. – why descriptions of results in this subchapter are suddenly for combined layers and not for each separately, as was the case in previous subchapters.

Reply: Thanks for your question. We have analyzed all the layers separately, and there were no significant conclusions for the individual layers. To facilitate statistical analysis of the data, the Danjiangkou Reservoir was divided into three layers: surface, middle and bottom.

Why is the Pearson corellation analysis (lines from 442) separate for point Q. On what basis the authors decided to present this analysis separately for this point, since the other analyzes for this point are not separated.

Reply: Thanks for your comments. The Q site was the entrance to the main canal of the Middle Route of the South-North Water Diversion Project. It was greatly affected by human water diversion activities, and the stratification was distinguished from other sampling sites. It was assumed that the WT and DO in the Danjiangkou Reservoir could be affected by adjusting the water diversion frequency of Q sampling site, thus affecting the vertical distribution of phytoplankton, so as to ensure water quality.

All these inaccuracies in the presentation of the results (once for layers, once for connected layers, once for a single point) make the results incomprehensible. They require redrafting, then the discussion will also have substantive value.

Reply: Thanks for your suggestion. We adjusted the order of the discussion section in 4.2 to be more coherent (lines 538-583). From general to specific, the relationship between the whole community and environmental factors was analyzed first, and then the relationship between the dominant species of the community and environmental factors was further analyzed, and finally the relationship between a specific phytoplankton and environmental factors was found.

General comments

There are some mistakes in the Latin names - in table 2 - it should be Ankistrodesmus instead of Ankistrodemus. The text says Aulacoseira granulate instead of Aulacoseira granulata (lines 438 and 475).

Reply: Thanks for your kind reminders. We corrected the error in the Latin names (lines 435, 448, and 472).

There are some mistakes in using spaces.

Reply: Sorry for this mistake. We checked the whitespace problem and corrected them.

Round 2

Reviewer 2 Report

After taking into account the comments, I believe that the article is very interesting for potential recipients. I have no more comments, thank you for considering my suggestions about the text. Thank you for the opportunity to review Your manuscript.